# MAKE IT EFFICIENT: DYNAMIC SPARSE ATTENTION FOR AUTOREGRESSIVE IMAGE GENERATION

## ABSTRACT

Autoregressive conditional image generation models have emerged as a dominant paradigm in text-to-image synthesis. These methods typically convert images into one-dimensional token sequences and leverage the self-attention mechanism, which has achieved remarkable success in natural language processing, to capture long-range dependencies, model global context, and ensure semantic coherence. However, excessively long contexts during inference lead to significant memory overhead caused by KV-cache and computational delays. To alleviate these challenges, we systematically analyze how global semantics, spatial layouts, and fine-grained textures are formed during inference, and propose a novel training-free context optimization method called Adaptive Dynamic Sparse Attention (ADSA). Conceptually, ADSA dynamically identifies *historical tokens crucial for maintaining local texture consistency* and *those essential for ensuring global semantic coherence*, thereby efficiently streamlining attention computation. Additionally, we introduce a dynamic KV-cache update mechanism tailored for ADSA, reducing GPU memory consumption during inference by approximately **50%**. Extensive qualitative and quantitative experiments demonstrate the effectiveness and superiority of our approach in terms of both generation quality and resource efficiency.

## 1 INTRODUCTION

Built upon a standard decoder-only autoregressive architecture, large language models (LLMs) (Su et al., 2024; Bai et al., 2023; Touvron et al., 2023a;b; Bi et al., 2024; OpenAI, 2023) generate text by sequentially predicting the most likely next token, achieving advanced language understanding and natural, human-like interactions. Inspired by this success, the autoregressive framework has been further extended beyond text, giving rise to powerful models capable of generating high-quality images and videos (Li et al., 2024a; Chang et al., 2022; 2023). These autoregressive models employ specially designed tokenizers (van den Oord et al., 2017; Tian et al., 2024; Chen et al., 2025; Qiu et al., 2025; Chen et al., 2024; Ma et al., 2025; Li et al., 2024b) to transform images into one-dimensional token sequences, adopting the same sequential probabilistic modeling approach used in text generation. This sophisticated process redefines visual content generation as a step-by-step token prediction task, where each visual patch is generated sequentially. Leveraging the strengths of self-attention, contextual learning, and cross-modal knowledge, this unified paradigm offers exceptional scalability and flexibility, enabling models to directly produce coherent, high-fidelity visual content from textual descriptions, thereby breaking new ground in cross-modal generation tasks.

However, the high computational cost of autoregressive models, especially when handling long sequences, poses a significant challenge. The quadratic complexity of conventional attention mechanisms leads to substantial memory consumption and increased computational overhead, limiting their scalability. To mitigate this issue, extensive research (Xiao et al., 2024a; Liu et al., 2025) has focused on efficient context computation techniques and KV-cache designs for LLMs, including sparse attention patterns, kernel-based approximations, and the replacement of attention layers with linear-complexity state-space models. While these methods can effectively reduce computational overhead, they often necessitate architectural modifications and model retraining, limiting their direct applicability to existing models. An alternative line of research has focused on enhancing inference efficiency by dynamically pruning redundant key-value vectors, thereby reducing memory consumption without altering the model architecture. However, these techniques have shown lim-

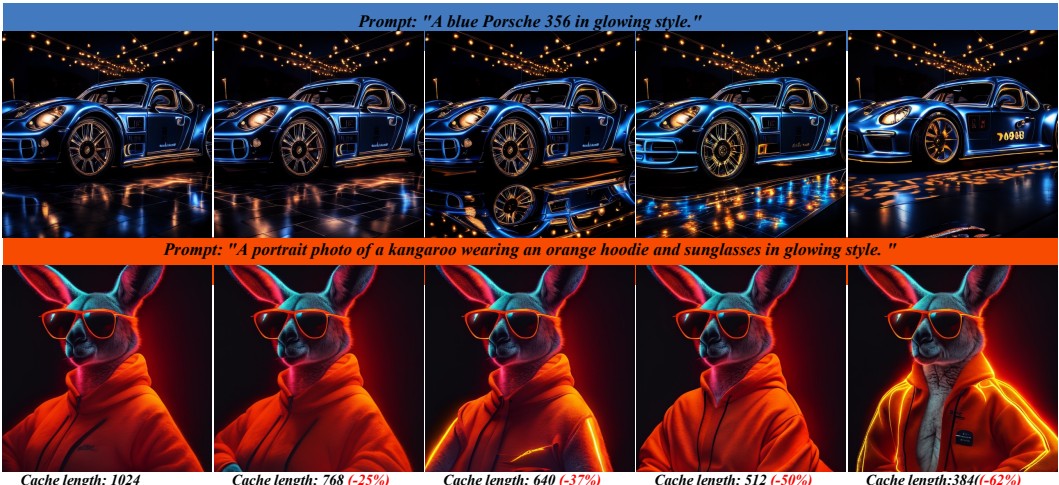

Figure 1: Achieving up to a 50% reduction in maximum context length during inference with our method. Samples are generated using LlamaGen, with the first column employing standard self-attention, while the remaining columns showcase the efficiency of dynamic sparse attention.

ited effectiveness in visual generation tasks. This limitation arises from the fundamental difference between text tokens and image tokens. Analyzing this from the perspective of information entropy per token, experimental results from Sparse Transformers (Child et al., 2019) show that for a $16 \times 16$ image patch, the total information content is approximately 26,291 bits. In contrast, in natural language processing (NLP) tasks, where the vocabulary size is $V = 65536$, the average information entropy of each token is $\log_2 65536 = 16$ bits. This stark disparity means that the information encapsulated within a single image token vastly exceeds that of a text token. Simply put, **while a single word can convey nearly complete semantic information, an image patch alone cannot provide a similar level of understanding.** This fundamental difference makes the direct application of text-based context optimization techniques to image generation inherently challenging.

Despite their inherently high entropy, image tokens exhibit strong spatial locality (He et al., 2024), with neighboring pixels frequently sharing similar visual characteristics. Empirical evidence, as illustrated in Fig. 2, further validates this observation, showing that a substantial portion of attention is consistently directed toward tokens positioned in the same column of the preceding row. This observation indicates that **not all tokens in the context hold equal importance**. While generating the current token, the model primarily relies on local tokens to accurately capture texture and details, while previous tokens mainly provide global layout and semantic context. Consequently, a locally constrained yet globally semantic-aware attention mechanism could significantly enhance both the efficiency and quality of autoregressive image generation. Motivated by these insights, we propose Adaptive Dynamic Sparse Attention (ADSA), a **training-free** strategy designed to significantly reduce the effective context length to minimize

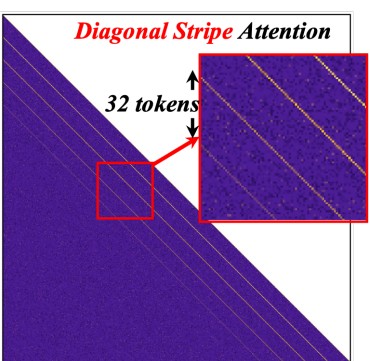

Figure 2: The attention scores of visual tokens in the LlamaGen-XL.

computational complexity in autoregressive models during inference. As illustrated in Fig. 3, ADSA retains the earliest image tokens to preserve global stylistic context while employing windowed attention to model local dependencies. It further adapts its attention patterns dynamically based on the information density of previously generated tokens. To further improve computational efficiency, we introduce **a dynamic KV-cache update strategy** that complements this sparse attention design. Unlike conventional approaches that maintain a full-length cache throughout inference, our method initializes the cache with only half the length and updates it adaptively during inference, significantly reducing GPU memory usage without compromising generation quality. Results in Fig. 1 demonstrate that by selectively attending to the most informative tokens, ADSA effectively reduces computational complexity while maintaining high-quality outputs.

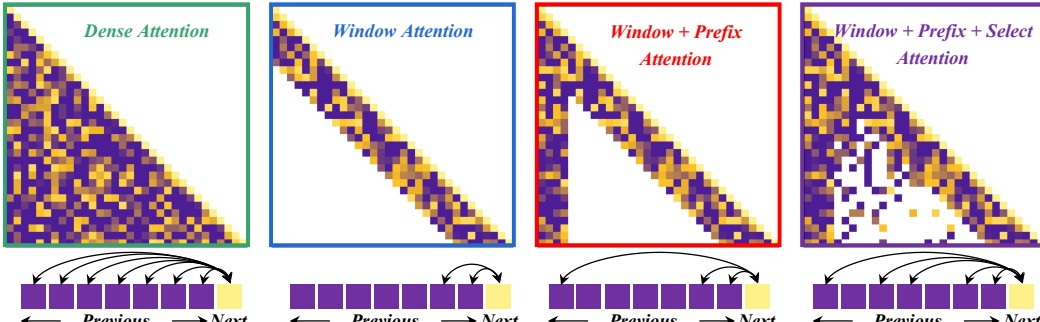

Figure 3: Dense Attention exhibits a time complexity of $O(T^2)$, with computational overhead increasing rapidly as the sequence length grows. Window Attention mitigates memory overhead by calculating key-value pairs for only the most recent $L$ tokens, providing efficient inference. However, its performance sharply degrades once the earliest tokens' keys and values are discarded. Window Attention with Prefix partially alleviates this issue by reconstructing the key-value states from the most recent $L$ tokens for each new token while preserving the influence of initial tokens. Adaptive Dynamic Sparse Attention dynamically adjusts the context during inference, selectively incorporating high-semantic-density image tokens, effectively mitigating the performance degradation.

## 2 RELATED WORKS

**Text-to-Image with Autoregressive Models.** Autoregressive text-to-image generation methods (Pang et al., 2024; He et al., 2025; Yu et al., 2025; Fan et al., 2024) reframe image synthesis as a next-token prediction process, generating images sequentially, token by token. These models employ a tokenizer to convert visual data into discrete tokens, which are then processed by a transformer using causal attention to maintain coherent image generation. Prominent methods, including VQGAN (Yu et al., 2022), DALL-E (Ramesh et al., 2021), and LlamaGen (Sun et al., 2024), leverage this framework by adopting GPT-style decoder-only architectures, effectively extending their text generation capabilities to visual synthesis. In contrast, some methods (Li et al., 2025) deviate from the standard raster order, opting for a random token generation strategy. This allows these models to simultaneously perform image synthesis and editing tasks, offering greater flexibility and control. By transforming two-dimensional images into one-dimensional token sequences, these models achieve strong text-image alignment. However, they often face limitations in the form of rigid generation orders and high computational costs, particularly when dealing with complex scenes.

**Efficient Context Computation in LLM.** Efficient context computation (Zhu et al., 2025; Gu et al., 2025) remains a critical and persistent challenge for large language models (LLMs), where models are typically trained on short contexts but are expected to maintain robust and consistent performance over significantly longer sequences during inference. To address this, state-of-the-art methods such as StreamingLLM (Xiao et al., 2024b) and LM-Infinite (Xiao et al., 2024a) have introduced a Λ-shaped attention window, enabling nearly unlimited input lengths by adaptively balancing global and local context focus. LongHeads (Lu et al., 2024) attempt to extend context through chunkwise retrieval from the middle cache. Other approaches, including MInference (Jiang et al., 2024) and RetrievalAttention (Liu et al., 2024), employ dynamic cache selection strategies to accelerate inference, yet they primarily enhance speed without directly addressing the challenge of robust context extrapolation. However, due to the fundamental differences between text and image modalities—where text tokens are compact and low in entropy, while image tokens are dense and high in entropy—these NLP-based strategies are not directly applicable to autoregressive image generation models. In contrast, we introduce **Adaptive Dynamic Sparse Attention (ADSA)**, a training-free, context-optimized attention mechanism specifically tailored for image tokens. ADSA dynamically reduces context length by selectively retaining the most informative tokens, effectively minimizing computational complexity while preserving global semantic consistency and local texture details. Notably, ADSA achieves these optimizations without the need for model retraining, making it a versatile and scalable solution for autoregressive image generation tasks.

## 3 ANALYSIS

To uncover the intrinsic content and structural control mechanisms of AR models, we conducted a systematic experimental analysis, focusing on their attention dynamics and sequential sensitivity.

### 3.1 HOW IS THE OVERALL STYLE FORMED?

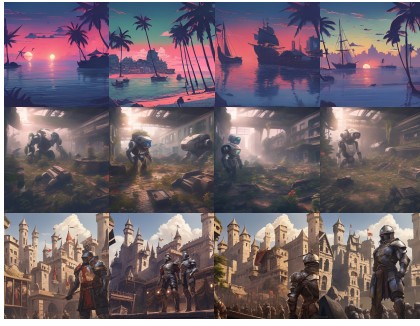

We conceptualize autoregressive continuous probabilistic modeling as a path exploration problem. We hypothesize that the tokens generated during the initial stages, despite their inherent high uncertainty, critically determine the trajectory of the image generation process, particularly influencing global style and color tone. This phenomenon arises fundamentally from the causal attention mechanism in transformer architectures, where early-generated tokens directly influence all subsequent token generations, thereby dominating the global structural and stylistic characteristics. In contrast, later-stage tokens primarily rely on local contextual dependencies, responsible for ensuring smooth color transitions and consistent local textures within individual image patches, with minimal

Figure 4: Early tokens define the global visual style and color palette.

impact on the overall image structure. This distinction emphasizes the significant role of early-stage tokens in establishing the global coherence and stylistic uniformity of generated images. To empirically validate this hypothesis, we conducted extensive experiments by generating images from a consistent textual prompt across multiple random seeds while methodically fixing the initial 5% of tokens. As shown in Fig. 4, the generated images consistently demonstrated highly similar global style and color tones, aligning well with our hypothesis. These observations strongly support our assertion regarding the decisive and consistent influence of initial-stage tokens on the final output.

### 3.2 HOW ARE THE FINE-GRAINED TEXTURES AND COLORS FORMED?

We observe that in autoregressive models like LlamaGen, tokens tend to assign higher attention weights to those in close proximity during attention computation. As illustrated in Fig. 2, the attention score assigned to a token generally decreases as the distance from the current token increases. This effect is particularly evident in the raster-order generation scheme, where each image token not only maintains a strong attention score with its immediately preceding token but also exhibits periodic local dependencies with tokens separated by a fixed interval. This behavior directly aligns with the two-dimensional spatial structure of images, where adjacent pixels along both horizontal and vertical axes demonstrate strong local correlations.

### 3.3 HOW ARE THE CONTENT CONSISTENCY AND CONTINUITY MAINTAINED?

Due to the inherent disparity in information density between image tokens and text tokens, the effectiveness of windowed attention varies significantly across modalities. As shown in Fig. 5, text generation typically benefits from a fixed-size attention window (e.g., 3 tokens), which is often sufficient to provide rich semantic context. Within such a window, the model can easily recover prior content—for example, the phrase "blue car" clearly indicates that a blue car has already been described, thereby anchoring the scene's primary object. In stark contrast, image generation faces a fundamentally different challenge. A fixed attention window containing 3 image tokens, each representing a small patch of pixels, conveys very limited visual information. Even if all patches contain predominantly blue pixels, the model cannot reliably infer whether these correspond to a blue car, a background region, or an unrelated object. Unlike text tokens—which are semantically discrete and inherently meaningful—image tokens are low-level and lack explicit semantic grounding. As a result, the model is constrained to enforcing only local coherence, such as consistent color and texture across neighboring regions, but remains incapable of capturing high-level structures or recognizing previously generated objects. This limitation frequently results in semantic drift, redundant generation, and incoherent scene composition.

Figure 5: Comparison of Information Density Between Text Tokens and Image Tokens in Window Attention. The figure illustrates the fundamental difference in semantic information density between text tokens and image tokens within a fixed attention window.

### 3.4 IS IT NECESSARY TO CACHE ALL KEY-VALUE PAIRS?

KV-cache substantially improves decoding efficiency in autoregressive models by avoiding redundant attention computations, reducing complexity from quadratic ($\mathcal{O}(T^2)$) to linear ($\mathcal{O}(T)$) with respect to sequence length $T$. However, complete caching of KV pairs significantly increases memory usage, causing sharp GPU memory overheads for long sequences. Existing multimodal understanding tasks typically prune visual tokens based on attention scores. In contrast, for autoregressive image generation, as illustrated in Fig. 2 and 5, tokens with high attention scores often cluster locally. Direct attention-based pruning thus risks weakening global semantic coherence, leading to repetitive generation and semantic degradation.

## 4 PROPOSED METHOD

### 4.1 ADAPTIVE DYNAMIC SPARSE ATTENTION

As discussed in Sections 3.1, 3.2, and 3.3, maintaining the overall consistency and coherence of generated images requires leveraging image tokens from multiple preceding stages as contextual references. To achieve this while preserving high generation quality and efficiently reducing context length, we propose Adaptive Dynamic Sparse Attention (ADSA), as shown in Fig. 6. Unlike conventional static sparse attention mechanisms used in large language models (LLMs), ADSA adopts an adaptive context selection strategy that dynamically adjusts based on the specific needs of each generation stage. This adaptive design enables the model to selectively focus on the most important tokens, ensuring both computational efficiency and superior image synthesis quality. Specifically, we define the long image input sequence as $I = \{I_t\}_{t=1}^{h \times w}$ where each token at time step $t$ is associated with a corresponding key $k_t$ and value $v_t$. Thus, the key-value cache

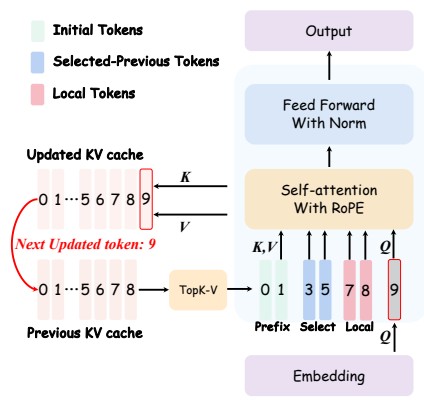

Figure 6: The Overview of Our Method.

(KV-cache) is defined as follows:$\mathcal{K}_{\text{cache}} = \{k_1, k_2, k_3, \ldots, k_t\}, \quad \mathcal{V}_{\text{cache}} = \{v_1, v_2, v_3, \ldots, v_t\}$. At each inference step $t$, we categorize the features stored in the KV-cache into three dynamically defined regions. The first $n$ tokens serve as the prefix, capturing the initial context that establishes the global style and semantic foundation of the image. Next, the most recent $m$ tokens closest to the current step $t$ are designated as the local region, ensuring fine-grained consistency and continuity in the generated content. The remaining tokens, located between the prefix and local regions, are classified as previous tokens, providing a broader contextual view. This process can be formally expressed as follows:

$$\mathcal{K}_{\text{cache}} = [\mathcal{K}_{\text{prefix}}, \mathcal{K}_{\text{Previous}}, \mathcal{K}_{\text{local}}], \quad \mathcal{V}_{\text{cache}} = [\mathcal{V}_{\text{prefix}}, \mathcal{V}_{\text{Previous}}, \mathcal{V}_{\text{local}}]. \quad (1)$$

Given that RoPE positional encoding is applied to the Q and K features, emphasizing positional dependencies, while the V features primarily capture the semantic content of tokens, we propose a TopK-V filtering method to efficiently reduce the length of previous tokens. Specifically, before

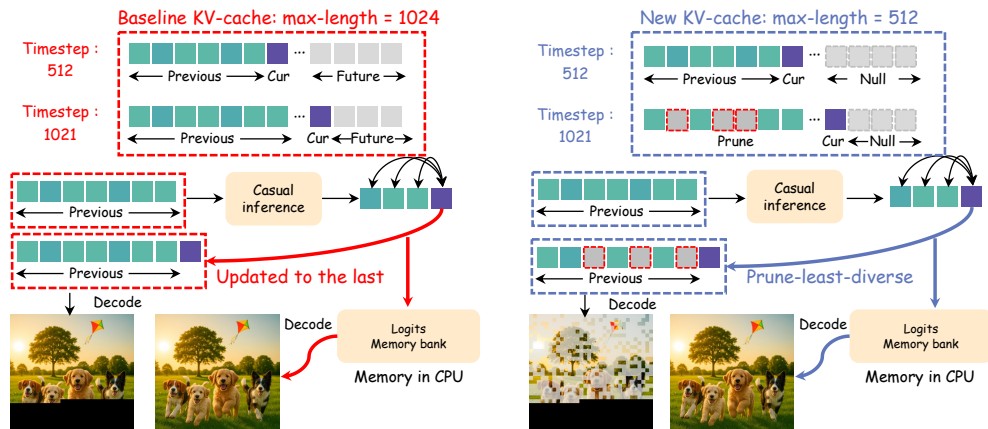

Figure 7: The Overview of Our Proposed Dynamic Sparse KV-Cache Updating Strategy.

computing attention, we calculate the cosine similarity among the V features in the KV-cache:

$$S_{ij} = \frac{v_i \cdot v_j}{\|v_i\| \|v_j\|}, \quad \text{for } v_i, v_j \in \mathcal{V}_{\text{previous}}, \ i \neq j, \quad S_{ii} = 0. \tag{2}$$

The average similarity score for each token is then calculated as:

$$S_i = \frac{1}{t-1} \sum_{j=1, j \neq i}^{t} S_{ij}. \tag{3}$$

We then identify the $K$ tokens with the lowest average similarity scores, ensuring semantic diversity among the retained tokens. Formally, this selection process is defined as:

$$\mathcal{I}_{\text{select}} = \text{argmin}_{\mathcal{I} \subseteq \{1,2,...,t\}, |\mathcal{I}|=K} \sum_{i \in \mathcal{I}} S_i. \tag{4}$$

Based on this selection, we obtain the filtered key and value sets:

$$\mathcal{K}_{\text{select}} = \{k_i : i \in \mathcal{I}_{\text{select}}\}, \quad \mathcal{V}_{\text{select}} = \{v_i : i \in \mathcal{I}_{\text{select}}\}. \tag{5}$$

This Top-K selection strategy ensures that the remaining tokens capture a wider and more diverse range of semantic information. By discarding the least similar tokens, the method introduces greater contextual diversity, enhancing the overall semantic richness. This adaptive filtering mechanism strikes an optimal balance between context length and semantic diversity, facilitating the generation of images that are both contextually coherent and rich in detail.

### 4.2 Dynamic Sparse KV-Cache Updating

As illustrated in Fig. 7, existing methods maintain a fixed-length key-value (KV) cache during inference, where the feature representations of newly generated image tokens are appended to the end of the cache at each step. The entire cache resides in GPU memory throughout the generation process, leading to considerable computational and memory overhead. In contrast, we introduce a more compact KV-cache mechanism that behaves identically to the baseline when the cache is not full. Once the cache reaches its capacity, we compute pairwise token similarity using Equations equation 2 and equation 3, and evict the most redundant token—i.e., the one most similar to others—before inserting the newly generated token. Meanwhile, all generated image tokens are offloaded to CPU memory during inference and only transferred back to the GPU for final image decoding, substantially reducing GPU memory consumption without compromising generation quality.

### 5 Experiments

To evaluate our method, we integrate it with the state-of-the-art autoregressive visual generation model, LlamaGen. For text-guided image generation, we generate 30,000 images and measure semantic alignment using CLIP scores (Radford et al., 2021) on the MS-COCO 2014 validation set with CLIP ViT-B/32. For class-conditional generation on ImageNet, we report Fréchet Inception Distance (FID) (Heusel et al., 2017) as the primary metric, alongside Inception Score (IS) (Salimans et al., 2016) and Precision/Recall to assess fidelity and diversity. All experiments were run on a single NVIDIA RTX 4090 GPU (48 GB).

Table 1: Quantitative evaluation on the ImageNet $256 \times 256$ benchmark.

| Models | FID↓ | IS↑ | Precision↑ | Recall↑ | KV Cache↓ | Context↓ |
|---|---|---|---|---|---|---|
| GigaGAN (Kang et al., 2023) | 3.45 | 225.5 | 0.84 | 0.61 | - | - |
| LDM-4 (Rombach et al., 2022) | 3.60 | 247.7 | - | - | - | 4096 |
| MaskGIT (Chang et al., 2022) | 6.18 | 182.1 | 0.80 | 0.51 | - | - |
| MaskGIT-re (Chang et al., 2022) | 4.02 | 355.6 | 0.80 | 0.51 | - | - |
| LlamaGen-XL (Sun et al., 2024) | 2.62 | 244.08 | 0.80 | 0.57 | 576 | 576 |
| ADSA-384 | **2.58** | 245.50 | 0.80 | 0.57 | 384 (-33.3%) | 384 (-33.3%) |
| ADSA-256 | 2.64 | **245.78** | 0.80 | 0.57 | **256** (-55.6%) | **256** (-55.6%) |

Table 2: Quantitative evaluation on the MS-COCO dataset.

| Models | CLIP Score↑ | KV Cache↓ | Context↓ |
|---|---|---|---|
| LlamaGen-XL (Sun et al., 2024) | **0.287** | 1024 | 1024 |
| ADSA-768 | **0.287** | 768 (-25%) | 768 (-25%) |
| ADSA-640 | **0.287** | 640 (-37.5%) | 640 (-37.5%) |
| ADSA-512 | 0.286 | **512** (-50%) | **512** (-50%) |

## 5.1 QUANTITATIVE RESULTS

**Class-conditional Image Generation.** In this subsection, we conduct a quantitative evaluation of class-conditional image generation using the LlamaGen-C2I-XL model, with a focus on the ImageNet $256 \times 256$ benchmark. In line with previous work, we generate images at a resolution of $384 \times 384$, resulting in a maximum context length of 576 during sampling, and subsequently resize them to $256 \times 256$ for evaluation. To assess the effectiveness of our method, we employ ADSA to selectively reduce the context to 384 and 256, respectively. As shown in Table 1, the ADSA-384 configuration achieves the best performance, even surpassing the baseline model with full context computation. ADSA-256 reduces the context length by more than half, resulting in only a slight increase of 0.02 in FID, while attaining the best performance in the IS metric.

**Text-conditional Image Generation.** In this subsection, we comprehensively evaluate text-conditional image generation using the LlamaGen-T2I-XL model on the widely-used MSCOCO dataset. Following prior work, we generate $512 \times 512$ images with a maximum context length of 1024. Leveraging ADSA, we progressively reduce the context length to 768, 640, and 512 tokens. As shown in Table 2, our method effectively reduces the context by selectively removing redundant tokens, while the CLIP scores of the generated images remain nearly unchanged, clearly demonstrating that our approach maintains strong semantic alignment with the given text prompts despite the substantially reduced context.

**GPU Memory-efficient Image Generation.** Our method substantially reduces GPU memory usage during autoregressive image generation by dynamically managing *KV-cache* updates, without compromising output quality. As shown in Fig. 9, when the batch size is small, memory consumption is dominated by model parameters. However, as the batch size increases, KV-cache becomes the primary bottleneck. Our approach achieves nearly **50% memory savings** on both the ImageNet and MS-COCO datasets, demonstrating strong generalization and scalability across diverse settings.

**Ablation.** To assess the contribution of the three distinct tokens in our method, we conducted a comprehensive ablation study. Specifically, we performed quantitative experiments on ImageNet using LlamaGen-C2I-XL, systematically remov-

Table 3: Results of ablation studies.

| prefix | select | local | FID↓ | IS↑ | Precision↑ | Recall↑ |
|---|---|---|---|---|---|---|
| × | ✓ | ✓ | 7.41 | 163.61 | 0.70 | 0.60 |
| ✓ | × | ✓ | 2.70 | 249.29 | 0.80 | 0.57 |
| ✓ | ✓ | × | 51.07 | 41.62 | 0.37 | 0.47 |
| ✓ | ✓ | ✓ | 2.58 | 245.50 | 0.80 | 0.57 |

ing each of the three tokens to evaluate their individual impact. As shown in Table 3, the complete ADSA method achieved the best performance. Notably, the largest performance drop occurred when the local token was removed, as the absence of local attention severely disrupted the locality of the image, leading to a substantial degradation in the high-frequency details of the generated images.

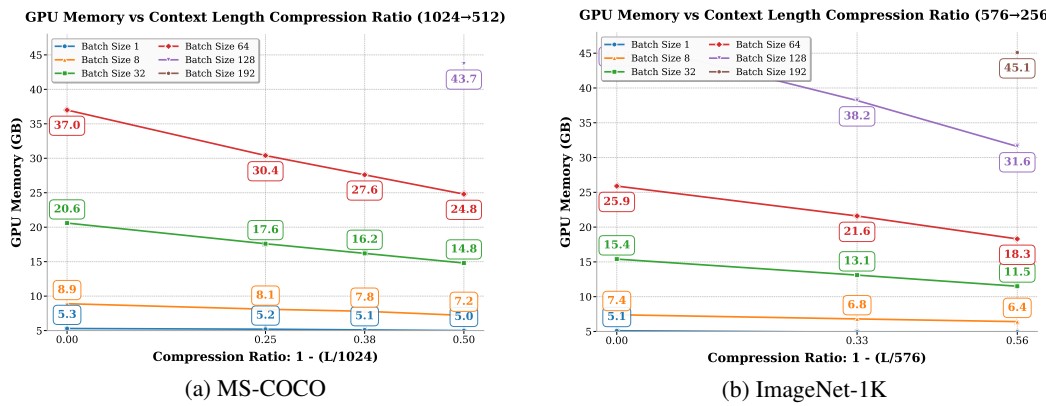

(a) MS-COCO

(b) ImageNet-1K

Figure 9: Shorter KV-cache lengths consistently reduce GPU memory usage across various datasets.

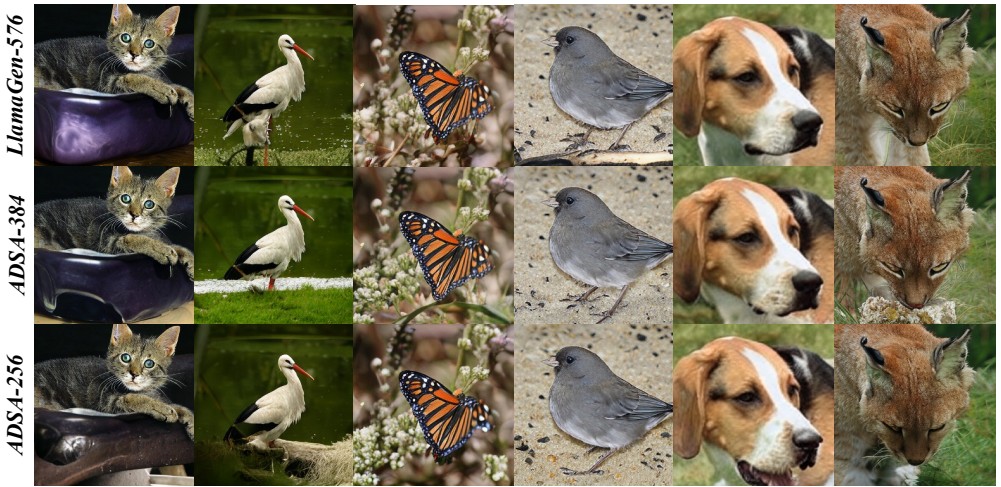

Figure 10: Samples generated by the LlamaGen-C2I-XL model using a next-token prediction paradigm under various dynamic sparse attention configurations.

## 5.2 QUALITATIVE VISUALIZATIONS

**Class-conditional Image Generation.** As shown in Fig. 10, our method generates high-quality images that seamlessly align with human cognition, preserving fine and intricate details even when the maximum context length is significantly reduced by half, given a specified generation category.

**Text-conditional Image Generation.** This subsection presents representative $512 \times 512$ image samples generated using our adaptive dynamic sparse attention mechanism. We examine the impact of reducing the maximum context length during inference from 1024 to 768, 640, and 512. As illustrated in Fig. 11, our method effectively reduces the context length for attention computation dur-

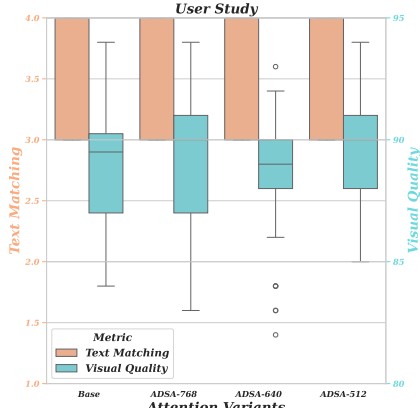

Figure 8: User study results.

ing inference, resulting in a significant decrease in memory consumption without any perceptible loss in image quality. Notably, we observed an unexpected yet intriguing phenomenon: as the context length is systematically and progressively shortened, the attention scores exhibit less smoothing from irrelevant tokens. This leads to a remarkable enhancement of high-frequency details in the generated images, contributing to a substantial improvement in their visual fidelity.

*"A peaceful Japanese Zen garden with a stone lantern, a koi pond, and cherry blossoms gently falling.*

*A tranquil snowy village at night, with warm light glowing from the cottage windows."*

*A majestic dragon soaring above snow-capped mountains, breathing a stream of blue fire.*

*"A magical forest with glowing mushrooms and fireflies, a crystal-clear stream winding through the trees."*

*Context length: 1024     Context length: 768     Context length: 640     Context length: 512*

Figure 11: Samples generated by the LlamaGen-T2I-XL model using a next-token prediction paradigm under various dynamic sparse attention configurations.

**User-Study.**    To evaluate the impact of our method on image quality, we conducted a user study with 48 GPT-generated text prompts guiding the LlamaGen-T2I-XL model. Ten users rated all the generated images. As shown in Fig. 8, all ADSA variant models performed well in text matching, effectively aligning the generated content with the descriptions. The visual quality of the images was consistently high, indicating their strong visual appeal.

## 6 CONCLUSION

In this paper, we introduce ADSA, a training-free sparse attention method that optimizes context usage during image generation, significantly reducing computational overhead without compromising image quality. ADSA exploits the visual structure of autoregressive models by dynamically evaluating token relevance and selectively computing attention. Experiments demonstrate that ADSA effectively halves the context length in LlamaGen, often improving generation quality. Future work will explore optimizing KV-cache management for further memory efficiency.

ETHICS STATEMENT

All authors have read and agree to abide by the ICLR Code of Ethics. This work does not involve interventions with human participants or personally identifiable information. We use only publicly available datasets under their original licenses and follow the terms of use. Potential risks and our mitigations are summarized below:

- **Privacy & Security.** We do not collect or release any personal data. When showing qualitative examples, all images/videos are from public datasets; any sensitive content is filtered.
- **Bias & Fairness.** We report results on multiple benchmarks and provide detailed settings to facilitate external auditing. We acknowledge possible dataset biases and encourage follow-up evaluation on broader demographics and domains.
- **Dual Use / Misuse.** The method could be misused to enable undesired large-scale labeling or surveillance. To reduce misuse, we release only research artifacts (code/configs) and exclude any tools for scraping or re-identifying individuals.
- **Legal Compliance.** We comply with licenses of all third-party assets (code, models, and datasets) and cite their sources. Any additional third-party terms are respected.
- **Research Integrity.** We document preprocessing, training recipes, and evaluation protocols; random seeds and hyperparameters are provided to enable reproducibility.

Where applicable, institutional review information is withheld for double-blind review and can be provided after acceptance.

REPRODUCIBILITY STATEMENT

We include training and evaluation details in the main paper and Appendix. Concretely: (i) all hyperparameters, optimization settings, and compute budgets; (ii) full data preprocessing and splits. Complete code and training logs will be open-sourced upon paper acceptance.

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

## A  MORE RESULTS

To better demonstrate the robustness of our model, we present additional experimental results as shown in Figure 12 and Figure 13.

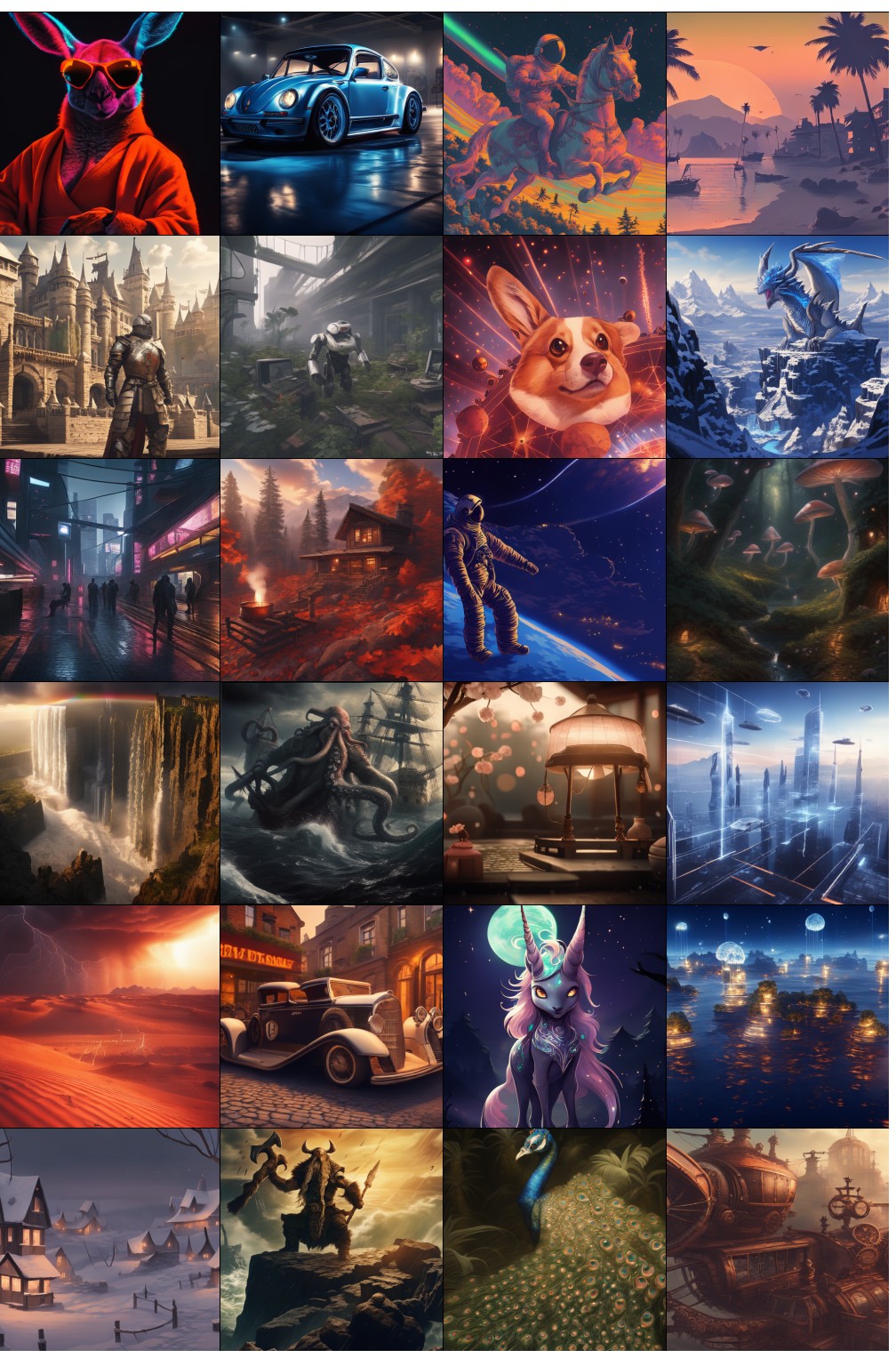

Figure 12: Text-conditional 512×512 image generation on ChatGPT-prompt.

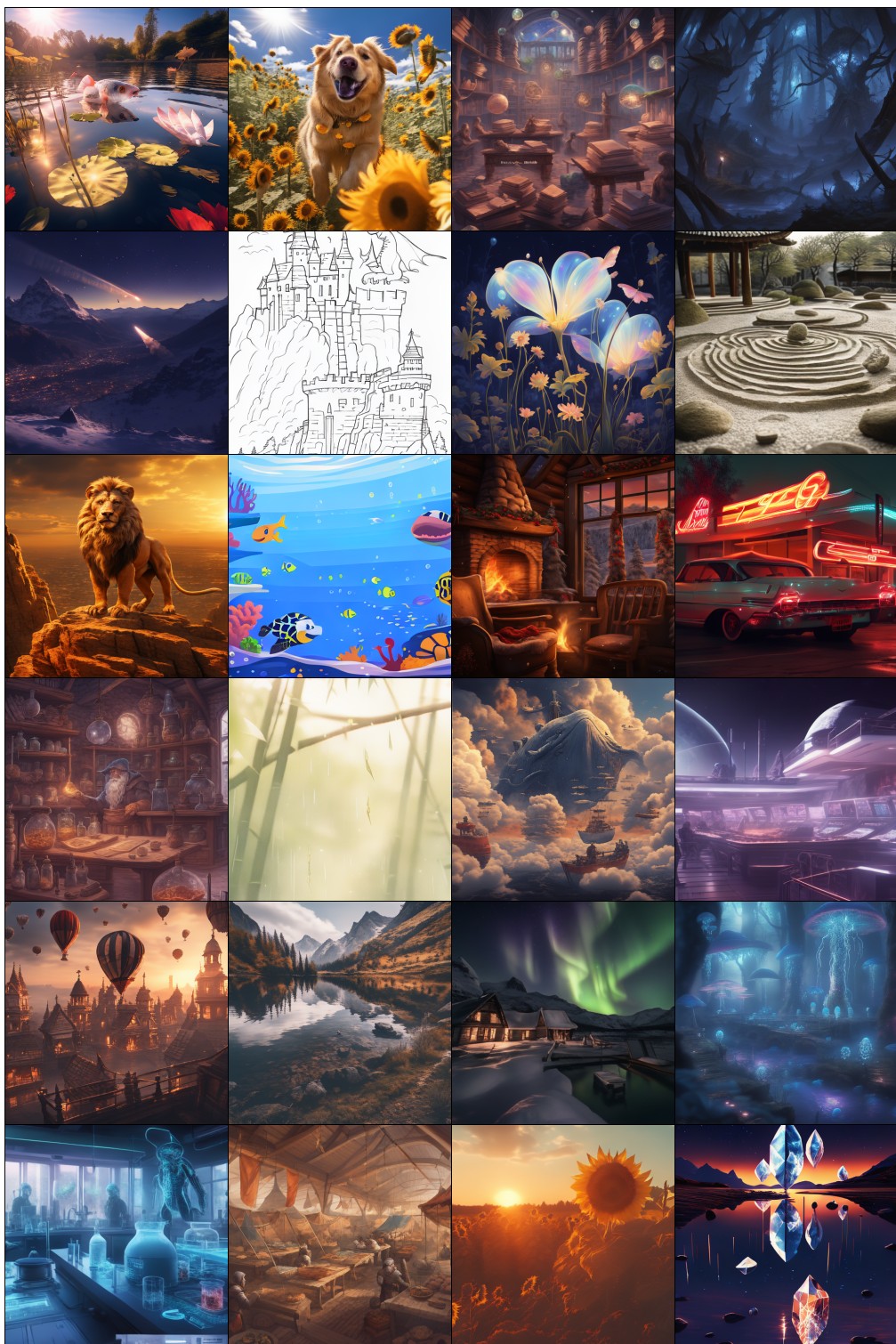

Figure 13: Text-conditional 512×512 image generation on ChatGPT-prompt.

## B  DISCLOSURE OF LARGE LANGUAGE MODEL (LLM) USAGE

In this paper, we used Large Language Models (LLMs) to assist in various aspects of the writing process. Specifically, LLMs were employed to help polish the writing, improve clarity, and enhance the overall presentation of the text. The models were utilized to provide suggestions for improving the grammar, coherence, and flow of certain sections of the manuscript. This assistance was integral to the refinement of the paper's language, but all scientific content, methodology, and conclusions were independently developed by the authors. The use of LLMs is limited to language-related tasks and does not extend to the intellectual contributions to the research findings or data analysis.

