# OpenReview forum: "Make It Efficient: Dynamic Sparse Attention for Autoregressive Image Generation"
_ICLR.cc/2026/Conference — ICLR 2026 Conference Withdrawn Submission_

### Official Review · Reviewer_G5KP · 2025-10-23

**Soundness:** 3
**Presentation:** 3
**Contribution:** 2
**Rating:** 2
**Confidence:** 4

**Summary:**

This submission proposes a training-free acceleration technique for auto-regressive text2img models. It focuses on shortening the K-V cache length through selection based on similarity and the combination with initial and local tokens.

**Strengths:**

- The motivation is clearly stated and convincing;

- The general idea is straightforward and reasonable;

**Weaknesses:**

Despite the idea being reasonable and straightforward, this work lacks both novelty and the evidence of its effectiveness:

- Insignificant Novelty: The selected K-V pairs have three parts: local tokens, sink tokens, and dynamically selected ones. The former two have been proposed earlier [1] and extensive works have studied similar topics [2,3,4].
    - More importantly, the only "new" technique should be the dynamic selection mechanism, because the other two are static and proposed by existing works. However, the ablation study in Tab. 3 shows that "select" brings very minor improvements.

- The purpose of this work is to accelerate the generation process. However,  the reported efficiency-related metrics are insufficient. Tab. 1&2 report the length of context, and Fig. 9 shows the relationship between the length and GPU memory. However, these results are not straightforward and insufficient. For example, both the memory usage and the generation latency should be reported directly in Tab. 1 & 2 to clearly show the trade-off between efficiency and generation quality.


[1] Efficient Streaming Language Models with Attention Sinks

2] Model Tells You What to Discard: Adaptive KV Cache Compression for LLMs

[3] ZipAR: Parallel Autoregressive Image Generation through Spatial Locality

[4] ZipVL: Efficient Large Vision-Language Models with Dynamic Token Sparsification

**Questions:**

- Is the method only compatible with LLaMaGen? This might affect its generality.

- How does the method perform compared to other paradigms such as NAR (Next-Scale Prediction) [5] or diffusion / flow-matching [6] models?

[5] Scalable Image Generation via Next-Scale Prediction

[6] FLUX

---

### Official Review · Reviewer_c7xK · 2025-11-02

**Soundness:** 2
**Presentation:** 3
**Contribution:** 1
**Rating:** 4
**Confidence:** 3

**Summary:**

This paper introduces ADSA, a training-free sparse attention method that optimizes context usage during image generation aiming to reducing computational overhead.

**Strengths:**

1. This paper is well-organized and easy to read.
2. The approach is very easy to follow.

**Weaknesses:**

1. The major concern of this paper is whether the proposed KV cache selection method can achieve obvious inference acceleration.
- For experiments, this paper only shows GPU memory usage compared with original entire context length methods. It seems that the GPU memory usage reduction is not obvious when batch size is small. However, small batch size inference is more common in real-world applications. Besides, the inference throughput is more important compared to GPU memory overhead (which is not included in the experiments).
- From theory, the actual algorithm complexity remains O(N^2) resulting from the cosine similarity calculation of the V features in the KV-cache. (as previous token >> local + prefix tokens). The introduced cosine similarity along with the argmin operation will lead to few throughput improvements.
2. The paper states that "inherent disparity in information density between image tokens and text tokens."  This leads to the inability to directly transfer the kv cache compression method on text (LLM) to image token generation. However, the approach proposed in this paper is actually very similar to the LLM kv cache compression method (such as Locret[1]), especially to some recent similarity-based kv cache compression methods in long video understanding such as FrameFusion[2]. This limits innovation on the one hand, and on the other hand, the lack of fair comparison with these works leads to insufficient effectiveness of the method.
- [1] Locret: Enhancing Eviction in Long-Context LLM Inference with Trained Retaining Heads on Consumer-Grade Devices
- [2] FrameFusion: Combining Similarity and Importance for Video Token Reduction on Large Vision Language Models
3. Some conclusions in Section 3 Analysis may be model-specific and lack persuasiveness and generalibility.
- Section 3.1 states that early tokens define the global visual style and color palette. The question is 1. Which image generation model do you use in Figure 4? And 2. As the image is generated in Raster order, the initial 5% of image tokens do not necessarily contain global information . Instead, they actually decide a very large patch of the generated images, thus controlling the style and palette.
- Section 3.3 states that "Text generation typically benefits from a fixed-size attention window (e.g., 3 tokens), which is often sufficient to provide rich semantic context." This claim is also not accurate.
4. The Last Question is about the compression rate. On one hand, more experiments with different compression rates should be included, on the other hand, as shown in Figure 2, maintaining 50% is too much than actually activated KV-Cache, from M-inference paper only 10% kv-cache is enough for approximate attention. What are the results of randomly selecting half kv cache? Should it still perform well.

**Questions:**

See weakness parts.

---

### Official Review · Reviewer_thWf · 2025-11-02

**Soundness:** 2
**Presentation:** 3
**Contribution:** 2
**Rating:** 2
**Confidence:** 4

**Summary:**

**Problem.** Autoregressive (AR) image generators suffer from quadratic attention cost, large KV-caches, and slow inference. **Method.** The paper proposes **Adaptive Dynamic Sparse Attention (ADSA)**: a training-free, inference-time context selection that keeps a prefix for global semantics, a local window for textures, and a small set of “previous” tokens chosen by TopK-V diversity; plus a **dynamic sparse KV-cache** that evicts redundant tokens and offloads to CPU. **Key innovations.** Token tri-partition with TopK-V selection (Eqs. 2–5), and a cache update/eviction scheme (Fig. 7). **Main results.** On ImageNet and COCO with LlamaGen-XL, ADSA reduces context/KV length by 25–50% with near-parity FID/IS/CLIP and ~50% GPU memory savings; qualitative samples claim unchanged or improved detail (Figs. 1, 10–11). **Significance.** If robust, this could be a drop-in speed/memory improvement for AR T2I without retraining.

**Strengths:**

- **Training-free, architecture-agnostic** drop-in idea with clear intuition (prefix for style, local for texture; Fig. 6).
- **Concrete formulation.** TopK-V selection by average V-similarity (Eqs. 2–5).
- **Reported memory relief.** Up to ~50% shorter cache/context with near-constant FID/IS/CLIP and memory curves across batch sizes (Tables 1–2; Fig. 9).
- **Simple ablation of prefix/local/previous** shows local window is critical (Table 3).

**Weaknesses:**

1) **Novelty vs. prior dynamic/sparse attention is insufficiently isolated.**
ADSA overlaps conceptually with Λ-shaped/window + selective cache ideas known in LLMs (e.g., StreamingLLM, LM-Infinite, LongHeads, Reattention, MInference, RetrievalAttention). The paper argues image tokens are high-entropy so NLP methods don’t transfer (Sec. 2), but lacks a *controlled* comparison where those baselines are adapted to LlamaGen and evaluated under the same protocol. Please add head-to-head against ZipAR/Neighboring AR and training-free infinite-context methods under identical settings (Sec. 2; Tables 1–2).

2) **Runtime and end-to-end latency are unreported.**
Tables show FID/IS/CLIP and memory, but not *wall-clock speed* (images/s, tokens/s) or latency distributions. The TopK-V step computes pairwise similarities within “previous” tokens (Eqs. 2–3), which adds non-trivial overhead; offloading to CPU (Fig. 7) can also stall on PCIe. Please report throughput, per-step latency (mean/p95/p99), and a breakdown of attention vs. selection vs. H2D/D2H copies (Eqs. 2–5; Fig. 7).

3) **Possible inconsistency in selection logic/text.**
Eq. (4) keeps tokens with *lowest* average similarity (least redundant), but the prose later says “discarding the least similar tokens,” which would remove diversity. Please clarify whether you **keep least similar / discard most similar**, and fix the affected sentences (Eqs. 2–5).

4) **Heuristic design lacks sensitivity analyses.**
Key hyperparameters (prefix length *n*, local window *m*, size of “previous” K, update cadence) are not tuned systematically. Add sweeps showing quality–memory–speed trade-offs, and robustness across prompts/classes (Fig. 6; Tables 1–3).

5) **Evaluation breadth is narrow.**
All results are on LlamaGen-XL; no evidence on other AR backbones/tokenizers (e.g., next-scale, non-VQ, randomized decoding). Provide at least one additional AR family to show generality (Sec. 5).

6) **COCO CLIP parity with 50% shrink is promising but fragile.**
CLIP difference at 1024→512 is ≤0.001 (Table 2). Please add confidence intervals across seeds and show DINO-based or TIFA-style metrics to mitigate CLIP-bias (Table 2).

7) **ImageNet results show tiny margins; statistical rigor is missing.**
ADSA-256 FID 2.64 vs baseline 2.62 (Δ=0.02) could be noise; IS slightly up. Provide μ±σ over ≥3 seeds and paired significance tests; report precision/recall curves, not single points (Table 1).

8) **User study methodology under-specified.**
Ten users, 48 prompts, but no details on randomization, rater agreement, or significance (Fig. 8). Include instructions, inter-rater reliability (κ), and CIs; release images and ballots (Sec. 5.2).

9) **“Early tokens define style” needs quantification.**
Fig. 4 qualitatively fixes first 5% tokens. Please quantify style/layout similarity as that percentage varies (e.g., LPIPS/Frechet color distance/SSIM over seeds), and test non-raster orders (Sec. 3.1).

10) **Failure modes & semantic drift.**
ADSA claims to preserve global semantics while pruning; show cases where aggressive pruning causes repetition/object loss, and report prompt categories most sensitive to K (Figs. 10–11).

11) **Reproducibility is deferred.**
Paper promises open-sourcing upon acceptance; for review, please provide configs (n, m, K, eviction policy), seeds, and minimal code to reproduce Table 1/2 numbers on a 4090 (Sec. 5, Repro. Statement).

12) **Claims of “often improving quality” need measurable evidence.**
The text speculates that shorter contexts “enhance high-frequency details” via less smoothing (Sec. 5.2). Validate with spatial-frequency/edge metrics or DISTS/LPIPS and human 2AFC on fidelity (Fig. 11 narrative).

**Questions:**

See Weakness

---

### Official Review · Reviewer_BE1H · 2025-11-09

**Soundness:** 3
**Presentation:** 3
**Contribution:** 2
**Rating:** 4
**Confidence:** 4

**Summary:**

This paper tackles the problem of excessive memory and computational overhead in autoregressive text-to-image generation caused by long attention contexts during inference.
It introduces Adaptive Dynamic Sparse Attention (ADSA), a training-free method that dynamically selects crucial historical tokens to preserve both local texture consistency and global semantic coherence.
ADSA reduces attention computation by focusing only on informative tokens, adapting its sparsity patterns based on token importance.  Extensive experiments confirm that ADSA achieves superior efficiency while maintaining image generation quality.

**Strengths:**

1. This paper addresses an important issue of computational efficiency in autoregressive image generation.

2. It provides insightful analyses of autoregressive generation mechanisms, highlighting the distinct roles of prefix, previous, and local tokens in shaping generated images (Section 3).

3. The proposed ADSA method effectively reduces the required context length while maintaining high-quality image generation.

**Weaknesses:**

1. The generality of the proposed ADSA is not fully explored. While its effectiveness is demonstrated on the LlamaGen model, further validation on other major autoregressive image generation models would strengthen the paper.

2. Although the paper presents solid analyses (Section 3) and quantitative results (Section 5), it remains unclear whether the proposed semantic-diversity-based context reduction strategy is optimal. An extended ablation study comparing it to simpler approaches (e.g., uniform sampling) and attention-based pruning (as mentioned in line 236) would be valuable.

3. The paper should further evaluate inference-time efficiency. Since ADSA adaptively selects and transfers contexts between GPU and CPU, this mechanism, while reducing GPU memory usage, may introduce additional onload/offload overhead that could negatively impact overall inference speed.

**Questions:**

None

---

### Note · Authors · 2025-11-18

I have read and agree with the venue's withdrawal policy on behalf of myself and my co-authors.